# Emphasis on Adipocyte Transformation: Anti-Inflammatory Agents to Prevent the Development of Cancer-Associated Adipocytes

**DOI:** 10.3390/cancers15020502

**Published:** 2023-01-13

**Authors:** Heeju Na, Yaechan Song, Han-Woong Lee

**Affiliations:** 1Department of Biochemistry, College of Life Science and Biotechnology, Yonsei University, Seoul 03722, Republic of Korea; 2Gemcro Corporation, Seoul 03722, Republic of Korea

**Keywords:** obesity and cancer, tumor microenvironment, cancer-associated adipocyte, inflammatory adipose tissue, adipocyte dedifferentiation, NSAID

## Abstract

**Simple Summary:**

Cancer cells that grow near adipose tissue inevitably exchange signals with adipocytes. The dynamic crosstalk between these two cell types facilitates the alteration of their cellular properties. The transformation of normal adipocytes into cancer-associated adipocytes (CAAs) provides a pro-tumorigenic niche for rapid tumor progression. However, the cancer-originated signals that mediate adipocyte transformation remain largely unknown. In this review, we discuss various inflammatory signals amplified in both obese and peritumoral adipose tissue. These inflammatory signals could mediate adipocyte transformation to phenotypes similar to CAAs by promoting adipocyte dedifferentiation and lipolysis. Epidemiological studies indicate a higher efficacy of nonsteroidal anti-inflammatory drugs in obese patients with cancer. Therefore, delivering anti-inflammatory agents can be a plausible therapeutic strategy to ameliorate the activation of tumor microenvironment components, especially adipocytes.

**Abstract:**

Of the various cell types in the tumor microenvironment (TME), adipocytes undergo a dynamic transformation when activated by neighboring cancer cells. Although these adipocytes, known as cancer-associated adipocytes (CAAs), have been reported to play a crucial role in tumor progression, the factors that mediate their transformation remain elusive. In this review, we discuss the hypothesis that inflammatory signals involving NF-ĸB activation can induce lipolysis and adipocyte dedifferentiation. This provides a mechanistic understanding of CAA formation and introduces the concept of preventing adipocyte transformation via anti-inflammatory agents. Indeed, epidemiological studies indicate a higher efficacy of nonsteroidal anti-inflammatory drugs (NSAIDs) in obese patients with cancer, suggesting that NSAIDs can modulate the TME. Inhibition of cyclooxygenase-2 (COX-2) and prostaglandin production leads to the suppression of inflammatory signals such as NF-ĸB. Thus, we suggest the use of NSAIDs in cancer patients with metabolic disorders to prevent the transformation of TME components. Moreover, throughout this review, we attempt to expand our knowledge of CAA transformation to improve the clinical feasibility of targeting CAAs.

## 1. Introduction

Adipocytes are the primary constituents of adipose tissue that store energy in the form of lipids. Adipocytes sustain tissue homeostasis by secreting diverse endocrine signals, including adipokines, lipids, and exosomes [1]. Under certain pathological conditions, adipocytes undergo dynamic alterations in their physical properties. For instance, obesity and diabetes accompany adipose tissue inflammation, which induces adipocyte hypertrophy and impaired metabolism [2]. Moreover, there is growing evidence that the interaction between adipocytes and cancer cells modulates adipocyte characteristics, causing them to secrete abnormal levels of adipokines and cytokines. However, only a few studies have suggested the mechanisms underlying adipocyte dedifferentiation [3].

Tumor cells secrete signals that transform nearby stromal cells to facilitate a favorable environment [4,5]. At the anatomical sites where cancer cells are adjacent to adipose tissue, cancer cells exert to activate adipocytes and generate cancer-associated adipocytes (CAAs). CAAs exhibit a dedifferentiated phenotype that entails a smaller size of lipid droplets. CAAs also possess curtailed mature adipocyte markers while acquiring fibroblast-like features. Cancer cells can induce the lipolytic activity of these adipocytes to exploit lipids as their main energy source [6]. Indeed, some cancer cells reprogram their metabolic dependency towards fatty acid oxidation to fully utilize long-chain fatty acids (FAs) [5,7]. To date, the pro-tumorigenic effects of CAAs have been well-defined (reviewed in [8]). However, only two cellular homeostatic pathways (Wnt and Notch) were suggested to generate CAAs, and only a few studies have attempted to provide a therapeutic strategy to prevent adipocyte transformation. [9,10].

Obesity-related cancers, including esophageal, gallbladder, colorectal, pancreatic, breast, endometrial, kidney, and thyroid cancer, have been reported to strongly correlate with patient body mass index (BMI) [11]. Obesity-related cancers tend to be proximal to adipose tissue, which includes subcutaneous, periprostatic, and visceral adipose tissue, indicating that the condition of adipose tissue can affect tumor progression [12]. A dramatic increase in the prevalence of obesity has encouraged many scientists to attempt to elucidate the link between obesity and cancer. However, despite intensive research, the underlying mechanisms of how obesity contributes to cancer progression remain unclear. In this review, we propose that both the tumor microenvironment (TME) and obese adipose tissue are overwhelmed by pro-inflammatory factors, and these inflammatory agents are sufficient for adipocyte transformation [13,14]. This supports the notion that anti-inflammatory agents could be used to modulate the TME, especially in patients with metabolic disorders.

Given that inflammation is one of the leading factors of tumorigenesis and tumor malignancy, anti-inflammatory agents have been evaluated for their anti-cancer effects [15]. Non-steroidal anti-inflammatory drugs (NSAIDs) target cyclooxygenase (COX) to inhibit prostaglandin (PG) production, which suppresses the sequential release of inflammatory mediators. Thus, regular use of NSAIDs has been suggested to inflict cancer-preventive effects [16]. However, according to multiple clinical practices, NSAID use as a chemotherapeutic agent has generated contradictory and inconsistent results [16]. Here, we emphasize the effect of NSAIDs on the inflammatory environment to suppress angiogenesis, fibrosis, and peritumoral adipocytes. Accordingly, preventing CAA transformation can be beneficial in cancers surrounded by a large population of adipocytes. This way, the chemotherapeutic approach simultaneously targets multiple TME components and potentiates NSAID use in patients with adipose tissue inflammation.

## 2. Characterization of CAA

The rapid expansion of solid tumors can drastically alter the local microenvironment, including adipose tissue. Such remodeling of the TME coincides with the increasing plasticity of its components, enabling angiogenesis, immune evasion, and cancer metastasis [4]. Since cancers actively exchange signals with TME components, how peritumoral adipocytes respond to those signals has garnered immense interest. 

CAAs are adipocytes with deeply modified properties via cancer-derived signals, and they are characterized by diminished lipid content, upregulation of genes associated with plasticity, and altered cytokine secretion [5,17,18]. Adipocytes grown with cancer cells using the in vitro cocultivation system underwent a significant loss of lipid content and showed an elongated morphology similar to that of fibroblasts [9]. Some studies have evaluated CAAs to have enhanced cellular plasticity and pluripotency, acquiring MSC-like features. In addition, upon cocultivation with breast cancer cells, CAAs display significantly low levels of CCAAT enhancer binding protein α (C/EBPα) and peroxisome proliferator-activated receptor γ (PPARγ), which are master regulators of adipogenesis and differentiation [19,20,21]. Such dedifferentiation of adipocytes enables them to migrate into the tumor mass and provides lipid metabolites to support tumor progression [7,21,22].

## 3. Oncogenic Roles of Transformed Adipocytes

Most remarkably, lipolytic enzymes in adipocytes are activated upon cocultivation with cancer cells [23]. Cancer cells are known to facilitate free fatty acids (FFAs) for energy production through fatty acid oxidation (FAO) when the available glucose is limited [6]. Such metabolic adaptations enable cancer cells to migrate and colonize other body parts while evading immune surveillance [24]. Therefore, inhibiting triacylglycerol (TG) breakdown and limiting lipid availability are used as chemotherapeutic strategies [25,26]. Due to an adipose tissue-rich environment, breast and prostate cancer tend to rely heavily on lipid metabolism [27,28,29]. These cancer cells curtail glucose consumption rates, increase fatty acid uptake, and overexpress the enzymes involved in β-oxidation [23,26,30]. Furthermore, FFAs can also be synthesized into lipid-signaling molecules, such as lysophosphatidic acid, sphingolipids, and prostaglandins. These signaling molecules are crucial messengers in the promotion of cell survival and proliferation [25,31]. The TME is characterized as metabolically restrictive, where CAAs supply metabolites, such as FFAs and lactate, to induce metabolic reprogramming of the TME components [32,33]. Promoting FAO-dependency of immune cells, CAAs construct TME favorable for rapid tumor growth and immune evasion [34,35].

Adipocytes secrete more than 600 metabolites, hormones, and cytokines, collectively known as adipokines [1]. Once activated by cancer cells, adipocytes secrete a wide array of adipokines, which promote tumorigenesis and tumor progression [36,37,38]. Leptin, autotaxin, and insulin-like growth factor (IGF) levels were upregulated in peritumoral adipocytes compared with those in distant mature adipocytes [38]. Patients with high-grade breast cancers exhibit enhanced levels of these adipokines, which leads to a worse prognosis [36,38,39,40]. In addition to breast cancers, these adipokines also promote the malignant behavior of colorectal cancer. The close relationship between obesity and colorectal cancers can be explained by the effect of differential adipokine secretion on cancer proliferation, metastasis, and resistance to therapy [37]. Unlike leptin, adiponectin level is downregulated in CAAs. Adiponectin is known to play anti-apoptotic, anti-inflammatory, anti-fibrotic, and insulin-sensitizing role, and low adiponectin levels support the rapid growth of cancer cells [41]. Obese patients tend to have a low adiponectin-to-leptin ratio, which can cause advanced tumors and a poor prognosis [42]. Other adipokines, such as lipocalin-2, IGF binding proteins (IGFBPs), and resistin, also promote cancer cell migration, invasion, proliferation, and resistance to therapy [22].

Similar to obese adipocytes, CAA-derived inflammatory factors include IL-1β, IL-6, TNFα, vascular endothelial growth factor (VEGF), C-C motif chemokine ligand 2 (CCL2), and matrix metalloproteinases (MMPs) [17,18,43]. Additionally, adipocytes co-cultivated with cancer cells exhibit a five-fold increase in the level of TNFα, which could participate in extracellular matrix (ECM) remodeling, angiogenesis, evasion of immune surveillance, epithelial-mesenchymal transition (EMT), and uncontrolled proliferation of cancer cells [43,44,45]. Secretory factors of CAAs also modulate immune cells within the TME by exerting immunosuppressive function during cancer development (reviewed in [17,34]). Overall, depending on the environmental conditions of adipocytes, their secretory factors can promote the malignant behavior of various cancer cells (Figure 1).

## 4. Wnt and Notch-Signaling to Induce CAA Transformation

Adipocyte dedifferentiation is a reverting process of mature adipocytes, which provides plasticity to transform adipocytes into fibroblast-like progenitor cells [7,9]. PPARγ and C/EBPα, the key regulators of adipogenesis and adipocyte differentiation, cooperatively provide the enzymes required for insulin sensitivity, lipogenesis, and lipolysis [46]. When affected by cancer cells, adipocytes undergo the downregulation of enzymes associated with adipogenesis, initiating adipocyte dedifferentiation (Figure 2). To date, two distinct pathways (Wnt and Notch) have been proposed to explain adipocyte dedifferentiation induced by cancer cells [10,47,48].

Cancer-mediated Wnt/β-catenin signaling in preadipocytes mainly suppresses adipogenesis [9,10,47]. Wnt3α and Wnt5α, the ligands of canonical Wnt signaling, are transferred to adipocytes during the cocultivation of cancer cells. Administration of Wnt3α and Wnt5α induces the c-Jun and activator protein-1 (AP1) signaling pathways while suppressing adipogenic signals [9,10]. Additionally, Li et al. reported that mechanical pressure could dedifferentiate adipocytes through Wnt signaling activation [47]. The rapid expansion of solid tumors results in intense physical stress on adjacent stromal cells. Adipocytes, under such stress, undergo dedifferentiation to acquire a mesenchymal stem cell (MSC) phenotype. Compression-induced dedifferentiated adipocytes (CiDAs) generated by mechanical pressure activate canonical Wnt/β-catenin signaling. The co-injection of cancer cells and CiDAs into mice resulted in the enhanced growth of cancer cells [47].

A study on liposarcoma (LPS) showed that adipocyte dedifferentiation could also be mediated by active Notch signaling. Herein, mice with constitutively active Notch1 were generated in an adipocyte-specific manner (Ad/N1ICD) [48]. The Ad/N1ICD adipocytes developed an impaired lipid metabolism pathway and underwent a loss of lipid messenger required for PPARγ activity. Owing to PPARγ ligand deficiency, these adipocytes underwent downregulation of fatty acid oxidation, lipid uptake, and aerobic respiration. This led to adipocyte dedifferentiation and exhibited an enriched human LPS gene signature. Conversely, PPARγ ligand supplementation and Notch inhibition re-differentiated adipocytes and suppressed the LPS transformation of mature adipocytes in mice [48].

Both Wnt and Notch signaling are closely linked with cellular development, differentiation, and homeostasis [49,50]. Furthermore, mounting evidence supports that Wnt and Notch signaling are also involved in the pathogenesis of chronic inflammatory diseases, including cancer-mediated inflammation [49,50,51,52]. Exposure to inflammatory cytokines, such as IL-6 and TNFɑ, could upregulate Wnt signaling and abrogate the differentiation of mature adipose cells [53]. Notch signaling could also be regulated by inflammatory cytokines such as IL-1β and TNFɑ [50,54]. Collectively, the driver of adipocyte transformation can also be regulated by diverse environmental conditions inflicting inflammatory signals on adipocytes.

## 5. Adipocyte Transformation via Cancer-Derived Inflammatory Factors

Tumor initiation and progression lead to local and systemic inflammation, which significantly disrupts tissue homeostasis [55]. In the TME, oncogenic and apoptotic signals recruit immune cells to the tumor expansion site [56,57]. In addition, enhanced ROS, hypoxia, and acidity in the TME present multiple inflammatory signals through cytokines, chemokines, growth factors, inflammasomes, exosomes, and metabolites [8,43,58]. Particularly in adipose tissue-rich environments, cancer cells attempt to fully utilize adipocytes through active crosstalk and maintain a favorable TME (Figure 3) [5,12,22,23,43,59].

The generation of CAAs can be mediated by inflammatory signals derived from cancer cells [60,61,62]. Inflammatory factors, such as transforming growth factor β (TGFβ) and TNFα, cooperatively inhibit the expression of genes associated with adipocyte maturation [60]. These molecules were found to be transported into adipose stromal cells at the invasive tumor front of the in vivo models. The transcriptional factors that maintain the adipocyte phenotype are downregulated in adipocytes cocultured with breast cancer cells [60]. Similarly, conditioned media (CM) from T47D breast cancer cells inhibited adipocyte differentiation. CM treatment downregulated the expression of C/EBPα, PPARγ, and adipocyte protein 2 (AP2); such phenomena could be abrogated using neutralizing antibodies against TNFα and IL-11 [61]. Thus, cancer-derived inflammatory cytokines, such as IL-11, TGFβ, and TNFα, can stimulate the reversion of mature adipocyte phenotypes.

Cancer cachexia is prevalent among patients with advanced cancer, and its symptoms involve severe loss of adipose tissue mass. Inflammatory cytokines, including TNF-α and IL-6, are responsible for cancer cachexia and are relevant to adipocyte transformation. Adipocytes exposed to such signals experience increased lipolysis, decreased lipogenesis, impaired lipid deposition, and browning [63]. Anti-IL-6 receptor antibodies could inhibit lipolysis and adipocyte browning in cachectic mice [62], and the depletion of IL-6 from tumor cells could prevent lipolysis in cocultivated adipocytes [64,65]. Additionally, cancer-derived IL-1β also triggers adipocyte transformation and adipose tissue cachexia [66].

Tumor cells also manipulate adipocytes to take full advantage when metastasizing into the adipocyte-rich bone marrow. Metastatic prostate carcinoma cells interact with the adipocytes in the bone to activate a pro-survival mechanism that allows rapid growth and escape from chemotherapy [67,68]. IL-1β secreted from cancer cells can sufficiently regulate the pro-inflammatory phenotype of adipocytes via the upregulation of COX-2 and monocyte chemoattractant protein-1 (MCP1). Since COX-2 activation increases prostaglandin E2 (PGE2) synthesis, modified adipocytes support cancer cells by promoting clonogenic growth and apoptosis resistance [69].

In cancer, the expression of inflammatory signals, such as TNFα, IL-1β, and IL-6, are crucial indicators of sensitivity to chemotherapy and patient survival. However, their effect on proximal adipose tissue has been overlooked [58,70,71]. As individual elements can alter adipocyte properties, it can be assumed that cancer-adipocyte proximity results in adipocyte transformation.

## 6. Adipocyte Transformation via NF-ĸB-Mediated Inflammation

NF-ĸB is a signaling hub for multiple inflammatory responses and is activated by various signals derived from solid tumors or impaired adipose tissues (Figure 4A). Active NF-ĸB mediates diverse signals to promote lipolysis and adipocyte dedifferentiation. In this review, we discuss the mechanisms by which NF-ĸB modulates adipocyte transformation and the attempts that have been made to suppress its activation.

In MSC, NF-ĸB expression increases upon early differentiation and is repressed by PPARγ during late adipogenesis. [72,73]. However, exposure to inflammatory signals, such as TNFα, IL-1β, TGFβ, or PGE2, activates NF-ĸB and halts terminal differentiation (Figure 4A) [74,75]. The active NF-ĸB binds to PPARγ and inhibits its interaction with the PPAR response element (PPRE) [76]. Inhibition of NF-κB using an IκBα repressor could prevent TNFα-mediated PPARγ loss and recover the adipogenic capacity of NIH-3T3L1 [77,78]. Simultaneously, NF-κB stimulates the expression of hormone-sensitive lipase (HSL) and induces lipolysis during inflammation. Intracellular translocation of NF-κB regulates the expression of perilipin1 (PLIN1) and hormone-sensitive lipase (HSL), which could be obstructed by NF-κB inhibitors [13].

NF-κB also mediates alternative pathways to induce adipocyte transformation [79,80,81,82,83,84,85]. Transcription factors regulated by NF-κB, such as TP53, c-MYC, and BCL2, repress the expression of PPARγ and C/EBP to reverse adipogenesis [85,86]. In addition, upregulated pro-inflammatory factors, such as CCL2, IL-6, IL-11, TNFɑ, IL-15, and PGE2, promote tissue inflammation to stimulate lipolysis and adipocyte dedifferentiation (Figure 4B) [79,80,81,82,83,84]. Alternatively, TNFα decreases PPARγ expression by promoting the caspase cascade pathway [79,87]. In NIH-3T3L1, active TNFR1 cleaves procaspase-8 into a functional caspase-8. The caspase activation cascade induces proteasomal degradation of PPARγ. Inhibition or genetic ablation of caspases could prevent TNFα-mediated PPARγ degradation [79,87].

The mitogen-activated protein kinases (MAPK) pathways are also involved in the inhibition of PPARγ and C/EBPs. High concentrations of C-C chemokine receptor type 2 (CCR2), TNFα, and IL-1α in obese patients activate p38 in adipocytes and inhibit C/EBPβ and PPARγ expression [88,89,90,91]. The pharmacological inhibition of MAPK stimulated adipocyte metabolism and adipogenesis. However, due to the lack of specificity, the effect of MAPK inhibitors on adipocytes should be evaluated more cautiously [91]. Another pro-inflammatory cytokine, IL-15, upregulates calcineurin to inhibit PPARγ and C/EBPα [92]. The calcineurin inhibitor FK-506 could restore PPARγ and C/EBP activity in the presence of IL-15 [92].

In terms of lipolysis, the cyclic AMP/protein kinase A (cAMP/PKA) pathway is well-recognized for phosphorylating adipose triglyceride lipase (ATGL), HSL, and perilipin1 (PLIN1) [93,94,95,96]. TNFα and PGE2 stimulate adenylyl cyclase and increase cAMP levels in adipocytes [94,97]. Genetic ablation of receptors for TNFα and PGE2 could prevent the activation of PKA and lipolysis; PKA inhibitors, H-89 and KT-5720, restored lipolysis and blood TG levels in obese mice [98]. IL-6 also stimulates lipolysis via the Janus kinase/signal transducer and activator of the transcription (JAK/STAT) pathway [81]. Extracellular vesicles (EV) derived from cancer cells contained IL-6, which was sufficient to induce HSL phosphorylation and delipidation in adipocytes. IL-6-neutralizing antibodies or STAT3 inhibitors could deactivate lipolysis by inhibiting HSL phosphorylation [81].

Collectively, NF-κB and its downstream pathways are crucial mediators of adipocyte transformation. Both direct and indirect deactivation of the NF-κB pathway can regulate lipolysis and adipocyte dedifferentiation. Therefore, we suggest that anti-inflammatory agents can effectively counter malignant TME formation by targeting adipocyte transformation.

## 7. Efficacy of Anti-Inflammatory Agents in Patients with Cancer

NSAIDs, such as aspirin, ibuprofen, mefenamic acid, celecoxib, piroxicam, sulindac, and diclofenac, act as blockers of the enzyme COX to inhibit PG synthesis [99]. Eicosanoids, the COX-derived PGs, are crucial mediators of inflammation. Exposure to PG derivatives, particularly PGE2, activates NF-κB to drive inflammation in adipocytes (Figure 5) [93]. Although COX is constitutively expressed in various cell types, COX-2 in cancer cells is thought to promote the malignant behavior of cancer cells [100,101].

COX-2 is overexpressed in various cancers, such as pancreatic, prostate, ovarian, breast, lung, and colon cancers; this overexpression stimulates angiogenesis, metastasis, and the chemotherapy resistance of the tumor [102]. Production of PGE2 is increased in cancer cells and can stimulate cancer cell proliferation and invasion [103,104]. Additionally, the exchange of pro-inflammatory signals between cancer cells and TME components sustains the repeated activation of the NF-κB and STAT3 pathways to exacerbate tumor malignancy [105,106]. For these reasons, NSAIDs are considered plausible candidates for cancer therapy and prevention. Furthermore, the long-term use of NSAIDs reduced the incidence of colorectal, esophageal, breast, and lung cancers [107]. As persistent inflammation is coupled with cancer progression, NSAID use has become a reasonable strategy for managing chronic inflammation and preventing the activation of TME components [108].

Continuous use of NSAIDs results in a lower incidence and mortality in patients with colorectal and lung cancers [109,110]. In addition, ibuprofen and piroxicam use significantly reduced the inflammatory potential in breast and colorectal cancers [111,112]. Moreover, patients with familial adenomatous polyposis (FAP) show decreased recurrence and lower polyp numbers when treated with sulindac [113,114]. The use of celecoxib in rat cancer models demonstrated a 90% tumor regression and a 25% reduction in the number of solid tumors [115]. Another study showed that ibuprofen inhibits cell proliferation in mouse and human colorectal cells. A 40–82% tumor regression and decreased tumor-induced angiogenesis were achieved by treatment with ibuprofen alone or in combination with the standard antineoplastic agents 5-fluorouracil or irinotecan [116]. Similarly, aspirin has demonstrated apoptotic and anti-proliferative effects in the HeLa cells. A synergistic anticancer effect for aspirin was observed when combined with doxorubicin, cooperatively inducing cell-cycle arrest, growth inhibition, and apoptosis in vitro and in vivo [117].

Numerous epidemiological studies support the notion that NSAID use benefits patients with cancer. A study involving 10,280 colorectal cancer cases showed a 27% reduction in colorectal cancer risk (odds ratio (OR) = 0.73; 95% confidence interval (CI): 0.54–0.99) [118]. A subsequent study investigated 2,118 women who had a female sibling with breast cancer. The use of non-COX-inhibiting NSAIDs did not correlate with reduced breast cancer risk among postmenopausal women. However, for premenopausal women, non-aspirin NSAIDs and aspirin reduced the risk by 34% (hazard ratio (HR) = 0.66; 95% CI: 0.50–0.87) and 43% (HR = 0.57; 95% CI: 0.33–0.98), respectively [119]. Another case-controlled study of 1,736 breast cancer patients in Spain reported a 24% reduction in breast cancer risk (OR = 0.76; 95% CI: 0.64–0.89) in those who use non-aspirin NSAIDs [120,121]. According to an investigation of 819 patients with prostate cancer, NSAID use significantly reduced prostate cancer risk (OR = 0.48, 95% CI: 0.28–0.79) [122]. Furthermore, an investigation of 7776 patients with ovarian cancer revealed that only aspirin use reduced the risk by 9% (OR = 0.91; 95% CI: 0.84–0.99) [123]. Other epidemiological data on NSAID use and the risk of pancreatic, prostate, bladder, and renal cancers remain controversial and limited. However, due to the conflicting consequences of the use of NSAIDs, it remains unclear whether they should be widely implemented against multiple types of cancer. In addition, epidemiological data imply that the anti-tumor effect of NSAIDs varies depending on their dose and duration, as well as the cancer types [124,125,126].

## 8. Anti-Inflammatory Agents in Obese Patients with Cancer

Some cancer types are surrounded by a large population of adipocytes, and they tend to rely on environmental cues when developing their malignant behavior. Obesity-related cancers can be greatly affected by the alteration of the TME during chemotherapy. Strikingly, NSAID use in obese patients resulted in a better prognosis after cancer treatment [127,128,129,130,131,132]. Particularly, patients with inflammatory adipose tissue may benefit from suppressing the chronic inflammation caused by peritumoral adipocytes. For instance, a daily dose of aspirin was more effective in patients with colorectal cancer who had a higher BMI (25–29 kg/m^2^ and >30 kg/m^2^). Unlike the somewhat increased risk with NSAID use in patients with normal weights (BMI < 25 kg/m^2^), obese individuals (BMI > 30 kg/m^2^) experienced a 56% reduction in risk upon regular NSAID use [129]. Similarly, in a case-controlled study involving 5,078 women, those who used NSAIDs regularly had a significantly lower risk of breast cancer incidence (OR = 0.78; 95% CI: 0.69–0.89) [127]. Another meta-analysis of 7,120 women with endometrial cancer showed that using aspirin more than once per week caused a 15% risk reduction among overweight and obese women (OR = 0.86; 95% CI: 0.76–0.98 and OR = 0.86; 95% CI: 0.76–0.97, respectively, for aspirin; OR = 0.87; 95% CI: 0.76–1.00 and OR = 0.84; 95% CI: 0.74–0.96, respectively, for non-aspirin NSAIDs) [131]. Interestingly, there was no correlation between aspirin use and cancer risk among women of normal weight.

Although NSAID use as a generalized chemotherapeutic strategy remains uncertain, specific populations with adipose-rich cancer exhibit positive outcomes. Clinical studies support that obese patients are likely to benefit more from the protective effects of NSAID use than normal-weight patients (Table 1). Crude measures, such as BMI and waist circumference, are insufficient to predict a positive response to NSAIDs. Thus, a more precise method should be considered to anticipate the effects of NSAIDs on the TME, especially in the case of an adipose-rich environment.

## 9. Future Perspective

To date, only Wnt and Notch signaling have been suggested as the major mediators of CAA transformation. Further studies are needed to identify the key drivers of the dynamic conversion of adipocytes. As many inflammatory signals are intertwined, it is necessary to comprehend the complex network of the corresponding pathways. To date, no study has specifically targeted inflammation to mitigate CAA transformation, even though the oncogenic role of CAA is starting to be recognized. We have shown that NF-κB could regulate adipocyte properties through multiple pathways (Figure 4). Although single delivery of an NF-κB inhibitor has distinct effects on the downstream signaling [133], it should be considered which inhibitor accounts for preventing CAA transformation.

NSAIDs target COX, which is the limiting enzyme in PGE2 synthesis. As PGE2 is a potent inducer of NF-κB, NSAID use also regulates multiple NF-κB downstream signals with minimal side effects (Figure 5). Numerous studies have demonstrated the inhibitory impact of NSAID use on cancer progression. In particular, we highlighted that patients with adipose-rich cancer benefited markedly from regular NSAID use. Furthermore, patients with a higher BMI showed better prognoses for breast, colorectal, and endometrial cancers (Table 1). It can be inferred that cancer patients with metabolic dysregulation or inflammatory adipose tissue may benefit from anti-inflammatory agents. This highlights the importance of evaluating an individual’s TME when predicting therapeutic outcomes and efficacy.

There remains a need to identify standard CAA markers through which cancer-derived cytokines drive CAA transformation. In addition, it is necessary to elucidate how adipocytes exposed to inflammatory signals share common and distinct features with CAAs. This will enable us to understand the resemblance between peritumoral and inflammatory adipose tissue, revealing the veiled connection between obesity and cancer. Furthermore, it is also essential to evaluate the condition of adipose tissue in patients with cancer to understand how the TME aids tumor growth. Currently, crude measures such as BMI and waist circumference are the only means used to evaluate adipose tissue condition; a more precise evaluation of patients’ TME status will improve the efficacy of chemotherapy, including that based on NSAIDs.

## 10. Conclusions

CAAs contribute to the secretion of inflammatory signals, metabolic reprogramming, and ECM remodeling in cancer cells. Despite the critical role of CAAs in the TME, the cell-intrinsic and extrinsic factors that trigger adipocyte transformation remain largely unknown. Inflammatory adipose tissues impose signals on adipocytes that are akin to the inflammatory factors secreted by tumor cells into the TME. These inflammatory signals have been found to be sufficient to modulate adipocyte properties (Figure 1). The major characteristics of CAA: suppression of adipogenic potential and activation of lipolysis, are found to be more pronounced in the adipose tissue of obese cancer patients. This implies that inflammatory adipose tissue provides more malignant TME through enhanced adipocyte transformation. Particularly, patients with impaired adipose tissue may significantly benefit from the delivery of anti-inflammatory agents via reconstructing the TME. NSAID use in obese patients yields better cancer prognosis, especially in those who bear tumors in an adipose-rich environment. This phenomenon may further explain the strong correlation between obesity and cancer progression.

## Figures and Tables

**Figure 1 cancers-15-00502-f001:**
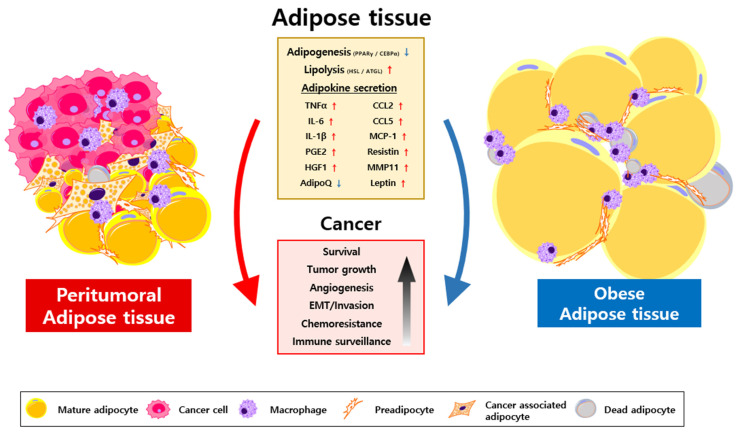
Oncogenic role of peritumoral and obese adipose tissue. Adipocyte characteristics are highly dependent on the environmental condition. At the invasive tumor front, adipocytes are exposed to various signals from cancer cells and are transformed into immature adipocytes. In parallel, under excessive consumption of nutrients, hypertrophic adipose tissue drives adipocyte death and inflammation. Both peritumoral and inflammatory adipose tissues exhibit downregulation of PPARγ and CEBPα expression, presenting the population of dedifferentiated adipocytes. Activation of lipolysis via key lipases, such as ATGL and HSL, leads to the secretion of various lipid metabolites that support tumor progression. Dedifferentiated adipocytes also provide an altered array of adipokines, which are considered pro-inflammatory and tumorigenic. The induction of adipocyte dedifferentiation and lipolysis contributes to cancer cell survival, growth, epithelial-mesenchymal transition (EMT), chemoresistance, and immune surveillance. Abbreviations: TNFα, tumor necrosis factor α; IL-6, interleukin-6; IL-1β, interleukin-1β; PGE2, prostaglandin E2; HGF1, human gingival fibroblast -1; AdipoQ, adiponectin; CCL2, C-C motif chemokine ligand 2; CCL5, C-C motif chemokine ligand 5; MCP-1, monocyte chemoattractant protein-1; Resistin, adipose tissue-specific secretory factor; MMP11, matrix metallopeptidase 11; EMT, epithelial-mesenchymal transition.

**Figure 2 cancers-15-00502-f002:**
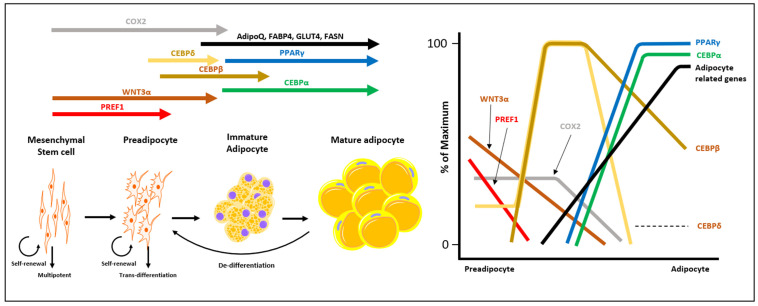
Alteration of gene expression during adipocyte differentiation. Differentially expressed genes throughout the adipocyte lineage are illustrated according to the timeframe. Mesenchymal stem cells exhibit the highest proliferative and pluripotent capacity, which decreases during adipocyte differentiation. Adipocyte dedifferentiation involves the loss of mature adipocyte markers, such as PPARγ and CEBPα, while acquiring the mesenchymal stem cell phenotype. Abbreviations: COX-2, cyclooxygenase-2; AdipoQ, adiponectin; FABP4, fatty acid binding protein 4; GLUT4, glucose transporter type 4; FASN, fatty acid synthase; CEBPα, CCAAT/enhancer binding protein α; CEBPβ, CCAAT/enhancer binding protein β; CEBPγ, CCAAT/enhancer binding protein γ; CEBPδ, CCAAT/enhancer binding protein δ; PPARγ, peroxisome proliferator-activated γ; WNT3α, Wnt family member 3A; PREF1, preadipocyte factor 1.

**Figure 3 cancers-15-00502-f003:**
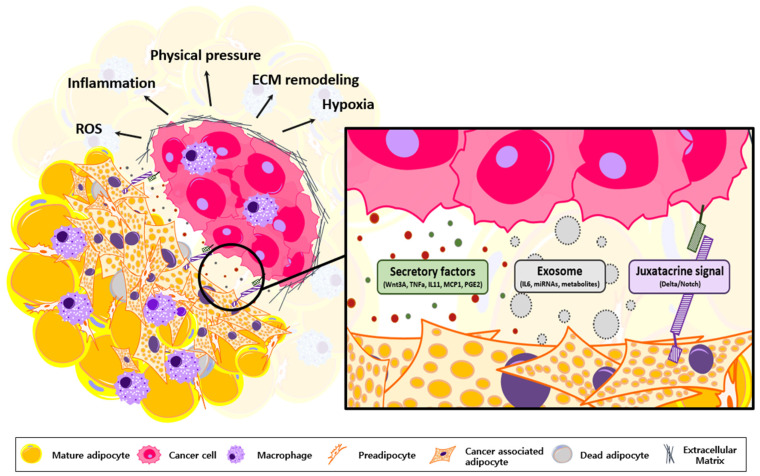
Generation of Cancer-Associated Adipocyte. Tumor cells exchange diverse signals with TME compartments to construct a favorable environment. In the adipose-rich environment, adipocytes undergo a transformation when influenced by cancer-derived secretory factors. Cancer cells and adipocytes also communicate via exosomes, which contain pro-inflammatory factors, miRNA, and metabolites. Juxtacrine signal also activates Notch signaling of adipocytes to initiate dedifferentiation. Abbreviation: TME, tumor microenvironment; ROS, reactive oxygen species; TNFɑ, tumor necrosis factor ɑ; IL-11, interleukin-11; MCP1, monocyte chemoattractant protein1; PGE2, prostaglandin E2.

**Figure 4 cancers-15-00502-f004:**
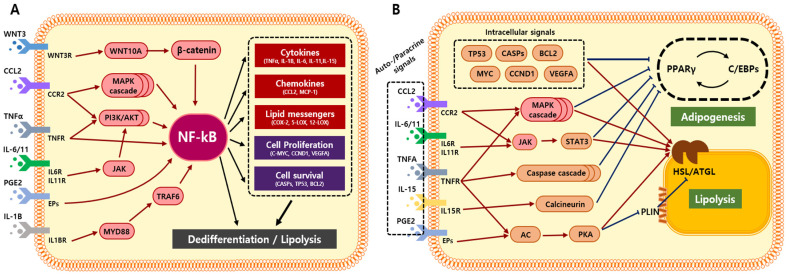
Summary of NF-κB mediated signal transduction regulating adipogenesis and lipolysis. (**A**) Pro-inflammatory signals activate NF-κB signaling to mediate adipocyte dedifferentiation and lipolysis. Secondary messengers are secreted to initiate an inflammatory signal cascade in adipocytes. (**B**) Autocrine and paracrine signals from NF-κB-activated adipocytes also regulate adipogenesis and lipolysis. Abbreviations: WNT3, Wnt family member 3; WNT3R, Wnt family member 3 receptor; WNT10A, Wnt family member 10A; CCL2, C–C motif chemokine ligand 2; CCR2, C-C chemokine receptor type 2; MAPK, Mitogen-activated protein kinases; PI3K, Phosphatidylinositol 3,4,5-trisphosphate kinase; AKT, protein kinase B; IL6, Interleukin 6; IL11, Interleukin 11; IL6R, Interleukin 6 receptor; IL11R, Interleukin 11 receptor; JAK, Janus family tyrosine kinase; IL1B, Interleukin 1B; MYD88, Myeloid differentiation primary response 88; TRAF6, Tumor necrosis factor receptor-associated factor 6; NF-κB, Nuclear factor kappa-light-chain-enhancer of activated B cells; IL15, Interleukin 15; MCP-1, monocyte chemoattractant protein-1; COX-2, cyclooxygenase-2; 5-LOX, 5-lipoxygenase; 12-LOX, 12-lipoxygenase; CCND1, Cyclin D1; VEGFA, vascular endothelial growth factor A; CASP, caspase; TP53, tumor protein p53; BCL2, B-cell lymphoma 2; JAK, Janus family tyrosine kinase; STAT3, Signal transducer and transcription 3; AC, Adenylyl cyclase; PKA, Protein kinase A; PPARΥ, Peroxisome proliferator-activated γ; CEBP, CCAAT/enhancer binding protein; ATGL, adipose triglyceride lipase; HSL, hormone-sensitive lipase; PLIN, Perilipin1.

**Figure 5 cancers-15-00502-f005:**
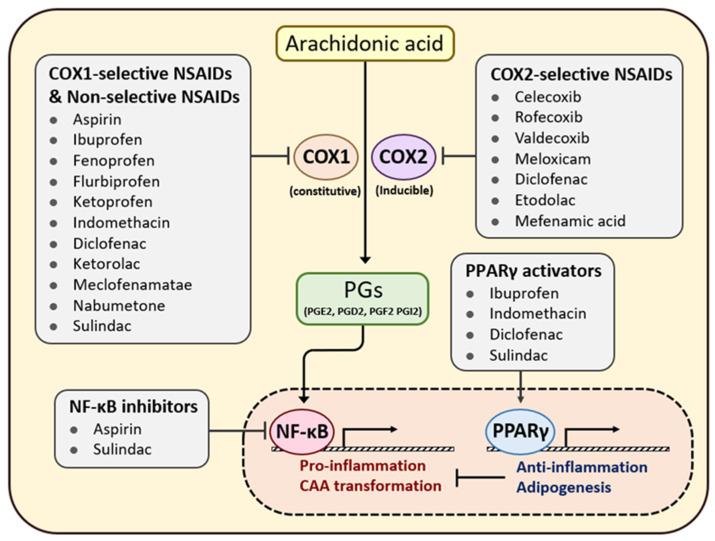
Suppression of NF-κB-mediated adipocyte transformation via NSAID treatment. COX is the major enzyme that catalyzes the conversion of arachidonic acid into prostaglandins. PGE2 and other prostaglandins induce adipocyte transformation and tissue inflammation, involving NF-κB signaling. To prevent systemic inflammation, NSAIDs are used to suppress COX-mediated prostaglandin production. Some NSAIDs alter adipocyte characteristics by non-selective inhibition of NF-κB and induction of PPARγ, which prevent adipocyte transformation by pro-inflammatory signals. Abbreviation: COX, cyclooxygenase; PGE2, prostaglandin E2; NF-κB, Nuclear factor kappa-light-chain-enhancer of activated B cells; NSAID, non-steroidal anti-inflammatory drug; PPARγ, Peroxisome proliferator-activated γ.

**Table 1 cancers-15-00502-t001:** The clinical results of NSAID use in obese patients with cancer Obesity-related cancers: esophageal, gallbladder, colorectal, pancreatic, postmenopausal breast, endometrial, kidney, and thyroid cancer. Abbreviations: NSAID, nonsteroidal anti-inflammatory drug; NA-NSAIDs; non-aspirin-NSAIDs: celecoxib, diclofenac, etodolac, fenoprofen, flurbiprofen, ibuprofen, indomethacin, ketoprofen, ketorolac, meclofenamate, mefenamic acid, nabumetone, naproxen/naproxen sodium, rofecoxib, sulindac, and valdecoxib; BMI, body mass index; OR, odds ratio; HR, hazard ratio; RR, risk ratio.

Cancer Type	Gender	NSAID	Frequency of Usage	BMI (kg/m^2^)	Number of Patients	Effect Compared to Normal	References
**Breast**	Female	Aspirin, ibuprofen, naproxen, indomethacin	Regular use	>25	5,078	Lower OR (22%)	[127]
**Breast**	Female	Aspirin, ibuprofen, celecoxib, naproxen, meloxicam	Daily	>30	440	Lower OR (52%)	[128]
**Colorectal**	Female, Male	Aspirin	Daily (325 mg)	>30	1,084	Lower RR (56%)	[129]
**Colorectal**	Female, Male	Aspirin	Daily (600 mg)	>30	54	Lower HR (10%)	[130]
**Endometrial**	Female	Aspirin, NA-NSAIDs	Over weekly use	>25, 30	87,189	Lower OR (15%)	[131]
**Endometrial**	Female	Aspirin	Regular use	>30	410	Lower OR (44%)	[132]

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
