# Peer review of "Emphasis on Adipocyte Transformation: Anti-Inflammatory Agents to Prevent the Development of Cancer-Associated Adipocytes"

_cancers, 2023, doi:10.3390/cancers15020502_

Round 1

Reviewer 1 Report

Comments on cancers-2119925

In this study, the author has studied “Emphasis on Adipocyte Transformation: Anti-Inflammatory Agents to Prevent the Development of Cancer-Associated Adipocytes.” A lot of studies have already been carried out on a similar topic, and comprehensive data is available in the literature. Sentence-making needs to be improved in this manuscript. The English language used in the manuscript needs major improvements as some punctuation and grammatical mistakes are present. Experimental designs required more clarity. Moreover, research models are not discussed in an understandable manner. Repetition of lines is common, which reflects that the author needs a more comprehensive way of thinking. It is obvious that the quality of the manuscript does not fulfill the standards of the journal, therefore, it should be reconsidered after major revision.

Specific comments:

1.      Page 1, line 25: “transformation remain elusive.” Please rephrase this.

2.      The Abstract needs to be critically revised, please add more information about the problem statement.

3.      Page 1, line 42-43: “In addition, adipocytes sustain tissue homeostasis by secreting diverse endocrine signals and cytokines.” Please add the names of some signals and cytokines secreted by adipocytes.

4.      Page 2, line 56-58: “Indeed, some cancer cells reprogram their metabolic de-56 pendency toward fatty acid oxidation to utilize abundant long-chain fatty acids (FA) [8, 10-13].” This is not an appropriate method to insert too many references in one place.

5.      Page 2: The whole introduction section looks for some improvement. Authors are advised to revise the introduction section carefully and add relevant data to support the problem statement and make a connection between each paragraph. The introduction needs more information between NSAIDs and CAAs. The introduction section also lacks information regarding the mechanism of NSAIDs to target CAAs. Overall, an introduction needs a major revision.

6.      Page 2: What is the research gap and novelty of the present study?

7.      The resolution of the figures is very good. Which software was used to draw the figures?

8.      Figure 2. “The authors are advised to add inflammatory factors involved in the transformation of CAA by Wnt and Notch signaling.”

9.      The authors are advised to make a table/figure representing the immune cells involved in CAAs regulations.

10.  Section 5 looks incomplete without an illustration.

11.  Please add the mechanism of action of NSAID drugs against cancer involving CAA transformation with an illustration.

12.  The authors have made little discussion about anti-inflammatory agents to prevent the development of CAAs. It is advised to add more data about anti-inflammatory agents with proper mechanisms and illustrations to catch the eyes of readers.

13.  It is advised to make conclusions in a single paragraph.

14.  It is suggested to add two more headings, such as “Crosstalk between cancer cells and adipocytes to modulate TME” and “Present challenges and future directions.”

15.  The authors are advised to add a list of abbreviations.

16.  Authors are advised to proofread the manuscript to overcome grammatical mistakes.

17.  Most of the references are outdated; please revise them and add updated data.

Reviewer 2 Report

The manuscript by Na et al. show that describes and depicts the various inflammatory signals amplified in both inflammatory and peritumoral adipose tissue. Additionally, this review discusses delivering anti-inflammatory agents can be a plausible therapeutic strategy to ameliorate the activation of tumor microenvironment components.

Cancer-associated adipocytes display a malignant phenotype and are shown at the invasive tumor front, which mediates the crosstalk network between adipocytes and cancer cells. It is better to discuss the mechanisms of adipocytes in the development of cancer, such as chemotherapy resistance, immune crosstalk and adipokines regulation.

I cannot find the figures in the manuscript.

Reviewer 3 Report

Heeju Na and colleagues discuss various inflammatory signals amplified in both inflammatory and peritumoral adipose tissue. These inflammatory signals could mediate adipocyte transformation by promoting adipocyte dedifferentiation and lipolysis, which exhibit similar phenotypes to CAAs. What’s more, epidemiological studies indicate a higher efficacy of nonsteroidal anti-inflammatory drugs (NSAIDs) in obese patients with cancer. 

Below are just some comments for the authors’ consideration.

1, No figures and tables in the manuscript. 

2, Authors discussed the benefit of different anti-inflammatory agents in obese patients with different type of cancer. Whether it is better to organize this information into a table.

Round 2

Reviewer 1 Report

The authors have carefully addressed all the comments. So, the manuscript should be accepted in the present form. 

Reviewer 2 Report

The authors addressed all the questions.